# Insights into the Use of Erythrocyte and Platelet Distribution Indices for Assessing the Extent of Coronary Lesions

**DOI:** 10.3390/medicina61111939

**Published:** 2025-10-29

**Authors:** Andrei-Catalin Zavragiu, Dumitru Sutoi, Oana-Raluca Radbea, Bogdan Chiu, Diana-Evelyne Mailat, Samuel Ardelean, Petre-Adrian Barzache, Ionut Dudau, Ovidiu-Alexandru Mederle, Minodora Andor

**Affiliations:** 1Doctoral School, “Victor Babes” University of Medicine and Pharmacy, Eftimie Murgu Square No. 2, 300041 Timisoara, Romania; andrei.zavragiu@umft.ro (A.-C.Z.); diana.mailat@umft.ro (D.-E.M.); samuel.ardelean@umft.ro (S.A.); adrian.barzache@umft.ro (P.-A.B.);; 2Department of Cardiology, Institute of Cardiovascular Diseases Timisoara, Str. Gheorghe Adam No. 13A, 300310 Timisoara, Romania; 3Department of Surgery, Emergency Discipline, “Victor Babes” University of Medicine and Pharmacy, Eftimie Murgu Square No. 2, 300041 Timisoara, Romania; mederle.ovidiu@umft.ro; 4Emergency Municipal Clinical Hospital, 300254 Timisoara, Romania; 5Faculty of General Medicine, “Victor Babes” University of Medicine and Pharmacy, Eftimie Murgu Square No. 2, 300041 Timisoara, Romania; raluradbea@gmail.com (O.-R.R.); bogdan.chiu@student.umft.ro (B.C.); 6Medical Semiology II Discipline, Internal Medicine Department, “Victor Babes” University of Medicine and Pharmacy, Eftimie Murgu Square No. 2, 300041 Timisoara, Romania; 7Department of Cardiology, ASCAR Clinical Hospital, Bulevardul Revoluției din 1989, No. 12, 300020 Timișoara, Romania

**Keywords:** red cell distribution width, platelet distribution width, mean platelet volume, Gensini score, coronary artery disease, chronic coronary syndrome, left ventricular ejection fraction

## Abstract

*Background/Objectives*: Red cell distribution width, platelet distribution width, and mean platelet volume are hematological indices derived from complete blood counts that have been increasingly investigated as potential cardiovascular biomarkers. This study aimed to assess the association between these indices and the extent of coronary artery disease, quantified by the Gensini score. *Methods*: We conducted a retrospective observational study of 240 patients hospitalized with typical angina who underwent elective coronary angiography at the Institute of Cardiovascular Diseases in Timișoara (January 2023–April 2024). Patients with hematological disorders, prior revascularization, or severe comorbidities were excluded. CAD severity was assessed by the Gensini score, with patients stratified into a low-score group (<50) and a high-score group (≥50). Laboratory and echocardiographic data were collected. Correlation analyses, ROC curve analysis, and multivariate regression were performed to evaluate predictors of CAD complexity. *Results*: Among 240 patients (81% male), 161 (67%) were in the high-score group. Compared with the low-score group, these patients had higher RDW (12.43 ± 0.68 vs. 12.70 ± 1.01, 95%CI: −0.52 to −0.02, *p* = 0.03), MPV (9.20 ± 0.88 vs. 9.45 ± 0.84, 95%CI: −0.48 to −0.02, *p* = 0.03), serum creatinine (*p* = 0.01), and potassium (*p* = 0.02), and lower ejection fraction (*p* < 0.001). RDW correlated positively with the Gensini score (rho = 0.28, 95%CI: 0.16–0.39, *p* = 0.001). Multivariate analysis identified RDW, MPV, and diastolic dysfunction as independent predictors. RDW showed weak discrimination for high Gensini burden (AUC = 0.57, 95% CI: 0.49–0.65, *p* > 0.05), consistent with a borderline, non-significant result. *Conclusions*: Elevated RDW and MPV are independently associated with greater coronary lesion burden. These routinely available indices may serve as novel, cost-effective markers for CAD risk stratification, warranting validation in prospective studies.

## 1. Introduction

Mean platelet volume (MPV), platelet distribution width (PDW), and red cell distribution width (RDW) are hematological parameters that can be easily obtained through routine laboratory testing, specifically during the performance of a complete blood count, using automated hematology analyzers. Although these indices are widely available and inexpensive to measure, they have gained considerable attention in recent decades due to their potential to reflect complex underlying pathophysiological processes, particularly those involved in cardiovascular diseases. An increasing number of clinical and observational studies have begun to explore these variables as potential markers of subclinical inflammation, platelet activation, endothelial dysfunction, and vascular remodeling [1,2,3,4,5,6].

Among these parameters, RDW, which quantifies the variation in erythrocyte volume within a blood sample, has been associated with a wide range of both chronic and acute conditions, including cardiovascular diseases, chronic kidney disease, systemic inflammatory disorders, and malignancies [7,8,9]. The scientific literature consistently reports that elevated RDW levels are correlated with a significantly higher risk of all-cause mortality, as well as an increased incidence of major cardiovascular events, such as acute myocardial infarction, heart failure, and stroke [10,11].

Proposed mechanisms by which RDW influences cardiovascular outcomes include chronic inflammation, oxidative stress, and malnutrition, which are factors that can impair erythropoiesis and reduce red blood cell lifespan [12].

Similarly, PDW, which is an index reflecting variability in platelet size, has emerged as a potential prognostic marker in several cardiovascular pathologies, including heart failure. Recent studies have shown that patients with congestive heart failure, particularly in its advanced stages, exhibit elevated PDW values, which are associated with adverse outcomes, including repeated hospitalizations and increased mortality [13].

Thrombosis, systemic inflammation, and platelet activation are key processes in the development and progression of atherosclerotic cardiovascular diseases. In this context, PDW, serving as an indicator of platelet heterogeneity, may reflect increased platelet reactivity and, consequently, a heightened risk of thrombus formation. Moreover, elevated PDW levels have been reported in numerous other conditions marked by systemic inflammation and vascular dysfunction, such as diabetes mellitus, metabolic syndrome, peripheral arterial disease, and most notably, ischemic heart disease [14,15].

Despite the growing body of evidence supporting the relevance of these hematological parameters in cardiovascular risk assessment, the exact pathophysiological mechanisms linking RDW and PDW to the severity and extent of coronary artery disease and to mortality remain only partially understood. It is hypothesized that low-grade chronic inflammation, persistent oxidative stress, iron metabolism disturbances, and bone marrow dysfunction may contribute to alterations in these indices [12]. However, further research is needed to clarify causal relationships and to determine whether these parameters are merely associated biomarkers or play an active role in the progression of atherosclerosis.

In this context, the primary objective of the present study is to specifically investigate the relationship between the severity of coronary lesions, which is quantified using the Gensini score (a validated angiographic tool for assessing coronary artery disease) [16], and elevated RDW, RDW-SD, PDW, and MPV values. Although the SYNTAX score is widely used in interventional cardiology to guide revascularization strategies [17,18], the Gensini score offers a more sensitive quantification of coronary atherosclerotic burden by integrating both the severity of stenosis and the functional importance of the affected coronary segment. The Gensini score was preferred because, in contrast to the SYNTAX score, it also incorporates subcritical (<50%) coronary plaques, thus providing a more comprehensive assessment of the overall atherosclerotic burden [16]. Unlike SYNTAX, which is primarily designed to assess lesion complexity for procedural decision-making, the Gensini score provides a continuous measure that better reflects the overall extent of coronary artery disease, making it more suitable for risk stratification and research on prognostic biomarkers [19]. Through this approach, we aim to identify significant correlations that may enhance our understanding of the prognostic value of these hematological indices and potentially support their integration into cardiovascular risk stratification strategies in routine clinical practice.

## 2. Materials and Methods

### 2.1. Study Design and Population

We conducted a retrospective observational study that included 440 consecutive patients who were hospitalized between January 2023 and April 2024 at the Institute of Cardiovascular Diseases in Timișoara.

The patients enrolled in our study cohort were hospitalized due to typical anginal symptoms for the purpose of undergoing elective coronary angiography. Typical angina was defined according to the European Society of Cardiology criteria as chest discomfort, often described as pressure, heaviness, or tightness, that is provoked by exertion or emotional stress and relieved by rest or nitroglycerin within a few minutes. Patients with known hematological conditions, such as hemolytic anemia, metastatic cancer involving the bone marrow, or those receiving iron supplementation, which could elevate plasma RDW and PDW levels, were excluded from this study. Patients were excluded from the study if they had a previous acute coronary syndrome or prior myocardial revascularization through coronary angioplasty with stent placement or aortocoronary bypass, since the Gensini score cannot accurately quantify atherosclerotic plaques in revascularized coronary segments. Patients who did not show angiographic evidence of coronary artery disease were also excluded, as the present study focused on evaluating the complexity of coronary plaques rather than simply diagnosing their presence. Thus, out of 420 initially evaluated patients, 200 were excluded: 137 had no angiographic evidence of coronary artery disease, 10 presented comorbidities that could influence the values of distribution indices, and 53 had a history of prior angioplasty (Figure 1).

The diagnosis of chronic coronary syndrome was based on the following criteria: typical chest pain lasting less than 20 min and resolving with rest, electrocardiographic changes observed in more than two concordant leads (specifically, ST-segment depression from the J point exceeding 0.2 mV in leads V1, V2, and V3, and over 0.1 mV in other leads), echocardiographic evidence of abnormal myocardial wall motion, a positive exercise stress test, or coronary CT angiography revealing significant coronary lesions. The study population was stratified according to coronary lesion complexity, as quantified by the Gensini score. Patients were assigned to the low-score group if they had a Gensini score below 50, while those with a Gensini score of 50 or higher were placed in the high-score group. The medical records of patients were systematically reviewed to collect data relevant to cardiovascular risk assessment, including the presence of diabetes mellitus, dyslipidemia, hypertension, smoking status, renal and hepatic function parameters, as well as a comprehensive blood count. The normal reference ranges for hematological indices were considered as follows: red cell distribution width of 11.5–14.5%, mean corpuscular volume of 80–100 fL, platelet distribution width of 9–17%, and mean platelet volume of 7.5–11.5 fL.

### 2.2. Angiographic Examination

Coronary angiography was conducted via radial or femoral access using 5 or 6 F Judkin catheters. Vascular access was obtained through the Seldinger technique. The coronary angiography procedures were jointly performed by two experienced interventional cardiologists. Each atherosclerotic plaque was carefully analyzed in multiple angiographic projections and, when appropriate, quantitatively assessed using the angiograph’s integrated measurement software. This collaborative approach ensured accurate evaluation of lesion severity and minimized potential errors associated with purely visual assessment. To calculate the Gensini score, each coronary artery lesion is assessed according to its degree of stenosis, or narrowing, and assigned a base score based on predefined intervals. Lesions exhibiting 1–25% stenosis are assigned a score of 1 point, while those with 26–50% stenosis receive 2 points. Lesions with 51–75% stenosis are scored at 4 points, and those with 76–90% stenosis receive 8 points. For stenosis levels between 91 and 99%, the score is 16 points; and complete occlusion, or 100% stenosis, receives the maximum score of 32 points. After determining the base score for each lesion, this score is multiplied by a weighting factor specific to the lesion’s location within the coronary artery tree. Due to its crucial role in cardiac blood supply, the left main coronary artery is assigned the highest multiplier, which is typically set at 5. Other significant segments, such as the proximal left anterior descending (LAD) and left circumflex arteries (LCXs), have multipliers of 2.5, while the mid-segment of the LAD is assigned a multiplier of 1.5. Distal segments of the LAD, LCX, and the right coronary artery receive a multiplier of 1. Following the application of these multipliers, the weighted scores of all lesions across the coronary segments are summed to yield the total Gensini score. This score provides a comprehensive assessment of coronary artery disease severity by integrating both the extent of stenosis and the functional importance of each affected coronary segment.

### 2.3. Statistical Analyses

Statistical analyses were carried out using JASP 0.19.1 software. The distribution of all continuous variables was assessed for normality using the Shapiro–Wilk test. Variables with a normal distribution were expressed as mean ± standard deviation and analyzed using parametric tests (Student’s or Welch’s *t*-test). Non-normally distributed variables were expressed as median (interquartile range, 25th–75th percentile) and analyzed using nonparametric tests, such as the Mann–Whitney U test, for comparisons between groups and Spearman’s rank correlation coefficient for assessing associations. Categorical variables were represented in terms of frequency (N) and percentage (%). Differences between proportions were assessed using the Chi-square test. The receiver operating characteristic curve for RDW values was generated to identify the optimal cut-off point in predicting the Gensini score. To assess the independent association of RDW, RDW-SD, mean corpuscular volume (MCV), MPV, and PDW with the Gensini score, potential confounding variables were initially tested in univariate regression analyses, and those with *p* < 0.05 were subsequently entered into the multivariate model. A *p* value of less than 0.05 was considered to indicate statistical significance.

## 3. Results

### 3.1. Baseline Clinical Characteristics

The study comprised 240 patients, of whom 195 were male (81%) and 45 were female (19%). The median age of the patients included in the study was 64 years (IQR 56–70 years), with a minimum age of 33 years and a maximum age of 87 years. Of the total cohort, 92 patients (38%) were smokers, 74 (30.83%) had diabetes mellitus, and 228 (95%) had hypertension. Additionally, 133 patients (55%) exhibited wall motion abnormalities, with a mean left ventricular ejection fraction of 48.96% (±7.12). The baseline characteristics of the study population are presented in Table 1. The study population was stratified into two groups according to the Gensini score, using a threshold of 50 to distinguish patients with simple coronary lesions from those with complex coronary lesions, as outlined in Table 2. The median age was significantly higher in the high-score group compared to the low-score group (65 IQR, 56–71 vs. 62 IQR 54–69 years, *p* = 0.04; Mann–Whitney U test). The Hodges–Lehmann estimate of the median difference was 3.0 years (95% CI: −6.0 to −0.2), indicating that older age was associated with more severe coronary atherosclerosis. The high-score group demonstrated a significantly greater proportion of patients with diastolic dysfunction (88% vs. 67%, 95%CI: 9.91% to 32.70%, *p* < 0.001) and a lower mean ejection fraction compared to the low-score group (47.71 ± 7.36 vs. 51.51 ± 5.84, 95%CI: 1.93–5.67, *p* < 0.001). No significant differences were observed between the groups regarding total cholesterol, low-density lipoprotein cholesterol, or triglyceride levels. The high-score group exhibited significantly higher serum creatinine (*p* = 0.008) and potassium levels (*p* = 0.02). Furthermore, among the assessed red blood cell and platelet distribution indices, both red cell distribution width (12.43 ± 0.68 vs. 12.70 ± 1.01, 95%CI: −0.52 to −0.02, *p* = 0.03) and mean platelet volume (9.20 ± 0.88 vs. 9.45 ± 0.84, 95%CI: −0.48 to −0.02, *p* = 0.03) were significantly elevated in the high-score group compared to the low-score group.

### 3.2. Correlation Analyses

A statistically significant but weak positive correlation (Table 3) was observed between the Gensini score and RDW (rho = 0.28, 95%CI: 0.16–0.39, *p* = 0.001) (Figure 2a), indicating that increased coronary lesion complexity is associated with greater anisocytosis. Although some comparisons reached statistical significance (*p* < 0.05), no meaningful correlation was observed between RDW–SD (rho = 0.14, 95%CI: 0.01–0.26, *p* = 0.02), MPV (rho = 0.15, 95%CI: 0.02–0.27, *p* = 0.01), and the Gensini score, as the corresponding Spearman’s rho values were below 0.25, indicating only very weak monotonic associations.

### 3.3. Receiver Operating Characteristic (ROC) Curve

To evaluate the predictive value of RDW for high Gensini score, we used an ROC curve and defined high Gensini scores as those above 50 points, which corresponded to approximately the 33rd percentile in our cohort. The optimal cut-off point of RDW was determined to be 12.7% with a sensitivity of 48% and a specificity of 66% for the whole cohort. Although RDW exhibited a numerically higher AUC than chance (0.57), the 95% CI: 0.49–0.65 encompassed 0.50 (*p* > 0.05), indicating borderline, non-significant discrimination and underscoring the exploratory nature of these findings. (Figure 3).

### 3.4. Univariate and Multivariate Regression

To investigate the correlation between MCV, RDW, RDW-SD, MPV, PDW, and the extension of coronary lesions, both univariate and multivariate linear regression analyses were performed, with the Gensini score as the dependent variable (Table 4). The univariate linear regression analysis revealed significant correlations between RDW, RDW-SD, MPV, ejection fraction, diastolic dysfunction, and wall motion abnormality with Gensini scores. Variables found to be statistically significant in the univariate regression analysis were subsequently entered into the multivariate regression model. RDW, MPV, and the presence of the diastolic dysfunction remained independently correlated with the Gensini scores (RDW: β = 16.44, 95%CI: 8.42–24.46, *p* < 0.001; MPV: β = 7.18, 95%CI: 1.24–13.11, *p* = 0.01; diastolic dysfunction: β = 16.63, 95%CI: 3.11–30.15, *p* = 0.01).

## 4. Discussion

Given that this topic is scarcely addressed in the current literature, the pathophysiological mechanisms underlying the association between variability in distribution indices and atherosclerotic plaque burden remain poorly understood. Elevated homocysteine levels are a well-established independent risk factor for cardiovascular diseases [20]. Homocysteine levels are influenced by vitamin B12 and folate levels, which, in turn, are inversely associated with the MCV of erythrocytes [21]. Therefore, it can be inferred that low levels of vitamin B12 and folate represent a risk factor for cardiovascular diseases, which is a conclusion supported by Liu et al. [22] in a study involving 8067 patients. Peng et al. [23] demonstrated in a study involving 344 individuals that RDW can serve as a predictor of homocysteine levels. Oxidative stress has been proposed as a factor influencing RDW but the relationship between oxidative stress biomarkers and RDW remains insufficiently characterized. Oxidative stress refers to a state of imbalance between oxidants and antioxidant defenses, where an excess of reactive oxygen species leads to oxidative damage to cells and tissues. Semba et al. [24] demonstrated an inverse relationship between serum antioxidant levels and RDW. White race, total serum carotenoids, α-tocopherol, and selenium were found to be significantly associated with lower RDW levels.

Larger platelets, due to their greater mass, exhibit higher metabolic and enzymatic activity compared to smaller platelets. They possess greater prothrombotic potential, characterized by elevated levels of intracellular thromboxane A2 and increased expression of procoagulant surface proteins [25]. Reactive oxygen species such as superoxide, generated during oxidative stress, contribute to increased platelet activation by stimulating intracellular calcium mobilization and impairing nitric oxide signaling. These mechanisms collectively facilitate a prothrombotic platelet response [26]. This prothrombotic state may account for the association between elevated PDW levels and the development of ST-segment elevation myocardial infarction in young patients, as reported by Cetin et al. [27]. They reported that each 1 fL increase in PDW levels was associated with a 13.5% higher likelihood of developing STEMI in young patients.

Our findings revealed a positive association between the severity of coronary artery disease and hematological indices such as RDW, indicating a possible relationship between the extent of atherosclerosis and changes in erythrocyte morphology. Moreover, patients with a higher coronary atherosclerotic burden, defined by a Gensini score > 50, exhibited significantly higher RDW and MPV values, suggesting a potential association between erythrocyte size variability, platelet morphology, and the severity of coronary artery disease. It is important to note that both PDW and MPV values remained within the normal range in both patient groups, which makes these differences more subtle and difficult to detect. This finding may lead to the establishment of new diagnostic thresholds for assessing coronary artery disease.

The present results corroborate earlier studies by demonstrating the significant association of RDW and MPV with the extent of coronary artery disease, as assessed by the Gensini score. Qiu et al. [1] reported a notable increase in Gensini scores among individuals with elevated mean platelet volume, with this association being particularly pronounced in the diabetic subgroup. Khalil et al. [6] demonstrated that diabetic patients with elevated coronary artery calcification scores and significant coronary artery disease exhibited higher red cell distribution width values. In contrast to the findings reported by Qiu and Khalil, who emphasized the relevance of diabetes mellitus in modulating hematological parameters such as RDW and MPV, our study did not specifically assess the presence or influence of diabetes in this context. Unlike previous studies in the literature, we chose to comprehensively assess all red blood cell and platelet distribution parameters, rather than focusing on individual markers; and to facilitate this analysis, the study population was stratified into two groups: one with simple coronary lesions and another with complex lesions.

The absence of a significant correlation between PDW and coronary lesion extent observed in our study is in line with the findings of De Luca et al. [28], who similarly reported no association between these variables. Cetin et al. [27] demonstrated that PDW and plateletcrit levels may serve as independent markers of STEMI in young patients, potentially reflecting an underlying prothrombotic state in this specific population. Unlike the previously cited study, which included patients with acute coronary syndrome, the present study evaluated individuals with chronic coronary artery disease. For this reason, the results may not be directly comparable. In contrast, Bekler et al. [29] reported no significant association between PDW and the severity of coronary artery lesions in patients with acute coronary syndrome, suggesting that PDW may serve as a predictor only in a limited subset of patients, rather than as a universal marker of coronary lesion complexity.

Tzur et al. [30] demonstrated in a study involving 1036 patients admitted to internal medicine wards that elevated PDW levels at the time of admission are associated with a more severe clinical profile and a higher risk of 90-day mortality. These findings suggest that PDW may serve as a useful prognostic marker for mortality risk in various clinical settings. However, in our study, which focused specifically on patients with chronic coronary artery disease, PDW did not show a predictive value for disease severity. It should be noted that short-term outcomes, such as 90-day mortality, were not evaluated in our cohort, which may partly explain the lack of prognostic significance. Further prospective studies are warranted to explore the potential predictive role of PDW in different clinical settings and over longer follow-up periods.

Recent evidence has shown that platelet populations are highly heterogeneous in terms of size, density, and functional reactivity, which may influence the interpretation of hematological indices such as MPV and PDW in cardiovascular disease. This heterogeneity may partly explain the modest and sometimes inconsistent associations between platelet indices and coronary atherosclerosis severity. In this regard, a recent study by Qiu et al. [31] identified distinct platelet subpopulations associated with disease severity and clinical outcomes, suggesting specific mechanisms of platelet disorders in COVID-19, sepsis, and systemic lupus erythematosus. These findings highlight the biological complexity of platelet activation pathways and provide a conceptual framework for understanding the variability of platelet-derived hematological biomarkers in cardiovascular settings.

A potential limitation of this study is the possibility of residual confounding related to medications and comorbidities that were not fully assessed. Certain conditions known to influence hematologic or inflammatory parameters, such as familial hypercholesterolemia [32,33], obstructive sleep apnea syndrome [34], atrial fibrillation [35], psoriasis [36], or vitamin D deficiency [37], may alter red blood cell morphology and platelet activation independently of coronary atherosclerosis. In addition, specific medications not evaluated in our cohort, including antituberculosis therapy [38] and agents such as doxycycline, azithromycin [39], or isotretinoin [39,40], have been reported to affect erythropoiesis, oxidative balance, or platelet indices. Although patients with major systemic and hematologic disorders were excluded, these unmeasured confounders may have contributed to subtle variations in RDW and MPV values and should be considered in future prospective studies.

Another limitation of this study is the relatively small sample size (n = 240), which may affect the generalizability of the findings. Additionally, the wide variability of the Gensini score, ranging from a minimum of 12 to a maximum of 209, posed challenges in establishing correlations with distribution indices such as RDW and MPV, whose values exhibit considerably less variation. Adapting or transforming the Gensini score into a narrower or standardized numerical scale might mitigate this issue in future studies. Another limitation is the absence of intravascular ultrasound imaging, which could have provided a more accurate quantification of the atherosclerotic burden within the coronary vessels.

## 5. Conclusions

Our findings demonstrate a significant positive correlation between the severity of coronary artery lesions and a routinely measured hematological parameter, which is the red cell distribution width. Specifically, patients with severe coronary lesions, defined by a Gensini score greater than 50, presented with higher levels of both RDW and MPV. This observation suggests a potential pathophysiological link between coronary atherosclerosis and morphological changes in red blood cells and platelets.

Taken together, our results support the clinical relevance of RDW and MPV as accessible, inexpensive, and non-invasive markers that may aid in the risk stratification of patients with chronic coronary syndrome. While these indices should not replace established diagnostic tools, they may serve as valuable adjuncts in identifying patients at higher risk and guide further investigation or therapeutic decision-making. Further prospective studies are warranted to explore the causal mechanisms underlying these associations and to validate the prognostic role of RDW and MPV in larger, more diverse populations.

## Figures and Tables

**Figure 1 medicina-61-01939-f001:**
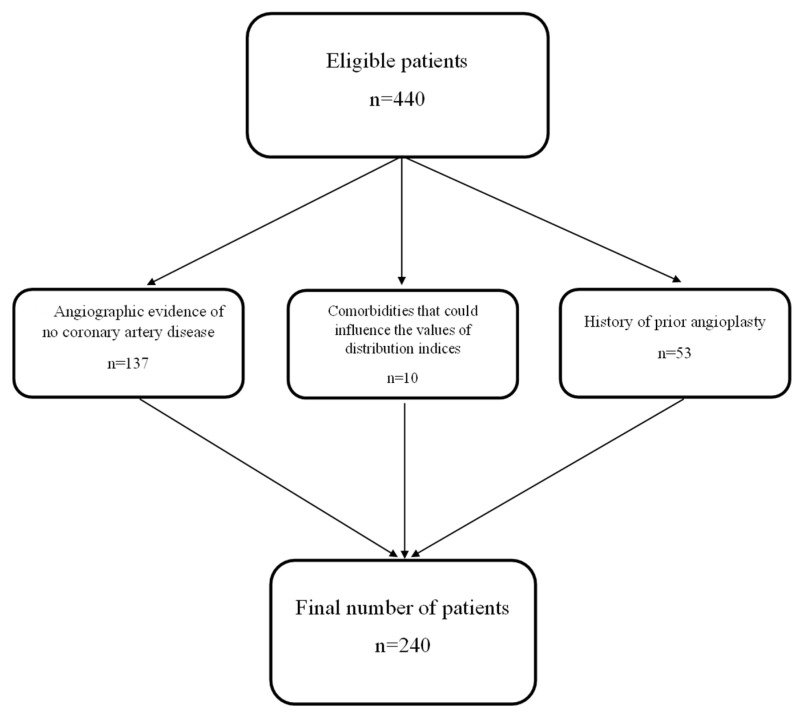
Flowchart.

**Figure 2 medicina-61-01939-f002:**
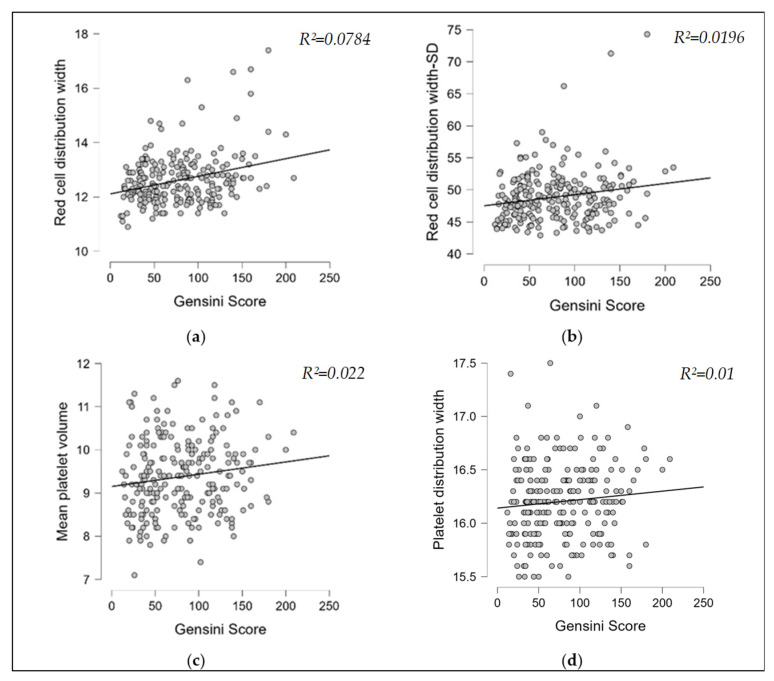
(**a**). Spearman’s correlation RDW–Gensini score; (**b**). Spearman’s correlation RDW-SD–Gensini score; (**c**). Spearman’s correlation MPV–Gensini score; (**d**). Spearman’s correlation PDW–Gensini score.

**Figure 3 medicina-61-01939-f003:**
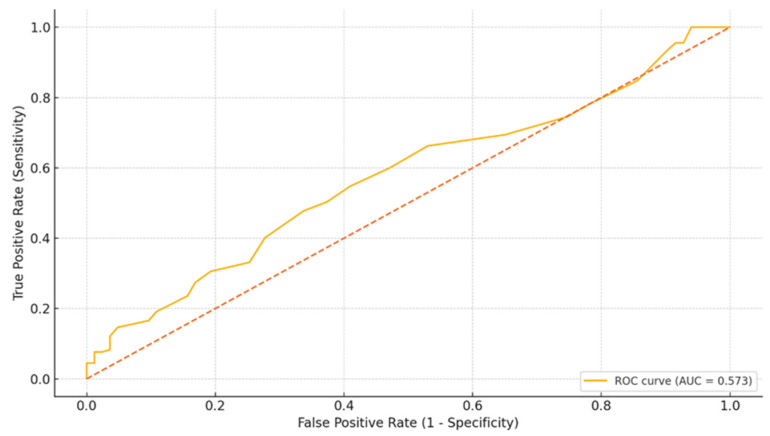
ROC curve of RDW for predicting severe coronary lesions (Gensini ≥ 50).

**Table 1 medicina-61-01939-t001:** Characteristics of study population.

Characteristics	N = 240
Male, % (N)	81% (195)
Female, % (N)	19% (45)
Smoker, % (N)	38% (92)
Diabetes mellitus, % (N)	30.83% (74)
Hypertension, % (N)	95% (228)
Age, years–median (IQR)	64 (56–70)
Gensini score–median (IQR)	75 (40–110)
Diastolic dysfunction, % (N)	81% (195)
Wall motion abnormality, % (N)	55% (133)
Hypokinesia, % (N)	29% (70)
Akinesia, % (N)	26% (63)
Left ventricular ejection fraction, %, mean ± SD	48.96 ± 7.12
TAPSE, cm, mean ± SD	2.18 ± 0.28
AST, U/L, mean ± SD	27.5 ± 12.2
ALT, U/L–median (IQR)	34 (27–46)
Creatinine, µmol/L–median (IQR)	90.17 (76.02–106.96)
Natrium, mmol/L, mean ± SD	139.73 ± 2.8
Potassium, mmol/L, mean ± SD	4.15 ± 0.45
Chloride, mmol/L, mean ± SD	101.89 ± 3.29
Fasting blood glucose, mmol/L–median (IQR)	6.1 (5.5–7.5)
Total cholesterol, mmol/L-median (IQR)	3.98 (3.21–4.60)
Low-density lipoprotein, mmol/L-median (IQR)	2.22 (1.60–2.77)
Triglyceride, mmol/L–median (IQR)	1.16 (0.86–1.63)
Hemoglobin, g/dL, mean ± SD	14.17 ± 1.5
Mean corpuscular volume, fL, mean ± SD	92.24 ± 4.97
Red cell distribution width, %, mean ± SD	12.61 ± 0.92
Red cell distribution width-SD, fL, mean ± SD	48.87 ± 4
Mean platelet volume, fL, mean ± SD	9.37 ± 0.86
Platelet distribution width, %, mean ± SD	16.2 ± 0.34

Note. Data were presented as either % (N), mean ± SD, or median [IQR 25–75%].

**Table 2 medicina-61-01939-t002:** Patient characteristics according to Gensini score.

Characteristics	Low Score < 50 N = 79	High Score ≥ 50 N = 161	*p* Value	CI (95%)
Male, % (N)	74% (59)	84% (136)	0.06	−0.58–21.68%
Smoker, % (N)	45% (36)	34% (56)	0.10	−1.95–23.92%
Age, years–median (IQR)	62 (54–69)	65 (56–71)	0.04	−6.0 to −0.2
Diabetes mellitus, % (N)	29% (23)	31% (51)	0.75	−10.66–13.60%
Hypertension, % (N)	94% (75)	95% (153)	0.74	−4.70–8.93%
Diastolic dysfunction, % (N)	67% (53)	88% (142)	<0.001 *	9.91–32.70%
Wall motion abnormality, % (N)	46% (37)	59% (96)	0.057	−0.37–25.82%
Left ventricular ejection fraction, %, mean ± SD	51.51 ± 5.84	47.71 ± 7.36	<0.001 *	1.93–5.67
TAPSE, cm, mean ± SD	2.20 ± 0.29	2.18 ± 0.27	0.61	−0.05–0.09
AST, U/L, mean ± SD	27.45 ± 12.39	27.52 ± 12.14	0.96	−3.37–3.24
ALT, U/L-median (IQR)	35 (28–45)	34 (27–46)	0.68	−3.00–5.00
Creatinine, µmol/L–median (IQR)	86 (73–103)	93 (79–119)	0.008	−15.02 to −1.76
Natrium, mmol/L, mean ± SD	139.74 ± 2.44	139.72 ± 2.97	0.95	−0.74–0.78
Potassium, mmol/L, mean ± SD	4.06 ± 0.37	4.19 ± 0.48	0.02 *	−0.23 to −0.01
Chloride, mmol/L, mean ± SD	101.94 ± 3.51	101.87 ± 3.19	0.86	−0.81–0.97
Fasting blood glucose, mmol/L–median (IQR)	6.17 (5.39–7.17)	6.11 (5.50–7.67)	0.64	−0.44–0.28
Total cholesterol, mmol/L-median (IQR)	4.09 (3.39–4.73)	3.88 (3.08–4.55)	0.10	−0.07–0.59
Low-density lipoprotein, mmol/L-median (IQR)	2.22 (1.68–2.85)	2.12 (1.60–2.74)	0.37	−0.13–0.36
Triglyceride, mmol/L-median (IQR)	1.17 (0.86–1.78)	1.15 (0.86–1.52)	0.42	−0.09–0.25
Hemoglobin, g/dL, mean ± SD	14.18 ± 1.35	14.17 ± 1.57	0.97	−0.40–0.41
Mean corpuscular volume, fL, mean ± SD	92.43 ± 4.85	92.14 ± 5.05	0.68	−1.06–1.63
Red cell distribution width,%, mean ± SD	12.43 ± 0.68	12.70 ± 1.01	0.03 *	−0.52 to −0.02
Red cell distribution width-SD, fL, mean ± SD	48.28 ± 3.03	49.16 ± 4.37	0.11	−1.95–0.20
Mean platelet volume, fL, mean ± SD	9.20 ± 0.88	9.45 ± 0.84	0.03 *	−0.48 to −0.02
Platelet distribution width, %, mean ± SD	16.16 ± 0.36	16.22 ± 0.33	0.20	−0.15–0.03

Note. Data were presented as either % (N), mean ± SD, or median [IQR 25–75%]. * = *p* value < 0.05 that means it is statistically significant

**Table 3 medicina-61-01939-t003:** Pearson’s correlation. (* *p* value < 0.05).

		Spearman’s Rho (CI 95%)	*p*
Gensini score	RDW	0.28 * (0.16–0.39)	0.001
	RDW-SD	0.14 * (0.01–0.26)	0.028
	MPV	0.15 * (0.02–0.27)	0.018
	PDW	0.10 (–0.03–0.22)	0.100
	MCV	−0.08 (–0.20–0.05)	0.212

**Table 4 medicina-61-01939-t004:** Univariate and multivariate regression. (* *p* value < 0.05).

Variable	Univariate β (CI 95%) *p*		Multivariate β (CI 95%) *p*	
MCV	−0.953 (−2.04–0.13)	0.087		
RDW	13.829 (8.20–19.45)	<0.001 *	16.44 (8.42–24.46)	<0.001 *
RDW-SD	1.996 (0.65–3.33)	0.004 *	−1.37 (−3.24–0.50)	0.15
MPV	6.988 (0.71–13.26)	0.029 *	7.18 (1.24–13.11)	0.01 *
PDW	11.926 (−3.66–27.51)	0.133		
Left ventricular ejection fraction	−1.336 (−2.08–−0.58)	<0.001 *	−0.852 (−1.77–0.06)	0.06
Diastolic dysfunction	20.441 (6.70–34.17)	0.004 *	16.63 (3.11–30.15)	0.01 *
Wall motion abnormality	13.801 (2.96–24.63)	0.013 *	0.06 (−12.94–13.07)	0.99

## Data Availability

All data supporting the findings of this study are contained within the article. Additional information can be obtained from the corresponding author upon reasonable request.

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
