# Peer review of "Insights into the Use of Erythrocyte and Platelet Distribution Indices for Assessing the Extent of Coronary Lesions"

_medicina, 2025, doi:10.3390/medicina61111939_

Round 1

Reviewer 1 Report

Comments and Suggestions for Authors

This study investigated whether routinely available blood count parameters including red cell distribution width, mean platelet volume, and platelet distribution width correlate with coronary artery disease severity as measured by the Gensini score. 
Major Revisions:
1.    The authors have overstated by using "Novel Insights" in the title;
2.    This is exploratory research showing weak statistical associations, the ROC analysis showed AUC=0.57 for RDW (barely better than chance);
3.    Need to include confidence intervals for all comparisons;
4.    In the discussion please mentionpotential confounding by medications and comorbidities not assessed;
5.    Should apply multiple testing correction (Bonferroni or FDR);
6.    Table 4 need to include confidence intervals for all estimates.
7.    The authors should consider reading  “Deciphering Abnormal Platelet Subpopulations in COVID-19, Sepsis and Systemic Lupus Erythematosus through Machine Learning and Single-Cell Transcriptomics" to provide insights into platelet heterogeneity that could inform the interpretation of hematological biomarkers in cardiovascular disease.

Minor:
1.    Remove redundant keywords; 
2.    Add R² values to scatter plots;
3.    Clarify definition of "typical angina";
4.    Add normal reference ranges for hematological parameters.

Author Response

Dear Reviewer, (PLEASE SEE THE ATTACHMENT)

We thank you for your insights for this manuscript. We are sure that these suggestions improve our article. 

Point-by-point response to Comments and Suggestions for Authors

Comments 1: The authors have overstated by using "Novel Insights" in the title.
Response 1: Thank you for pointing this out. Therefore, we have changed the title
to ”Insights into the Use of Erythrocyte and Platelet Distribution Indices for Assessing the
Extent of Coronary Lesions”

Comments 2: This is exploratory research showing weak statistical associations, the ROC
analysis showed AUC=0.57 for RDW (barely better than chance).
Response 2: We agree that the study is exploratory and that the ROC analysis for RDW
showed only a modest discriminative performance. The text has been revised accordingly to
better reflect this limitation. The test was re-performed and the explanation was revised to
accurately reflect the present findings.
“[RDW showed weak discrimination for high Gensini burden (AUC = 0.57, 95% CI: 0.49–0.65, p >
0.05), consistent with a borderline, non-significant result.]” page 1, line 42-43
“[Although RDW exhibited a numerically higher AUC than chance (0.57), the 95%CI:0.49–0.65
encompassed 0.50 (p > 0.05), indicating borderline, non-significant discrimination and underscoring
the exploratory nature of these findings.]”page 8, line 241-243

Comments 3: Need to include confidence intervals for all comparisons
Response 3: We thank the reviewer for this important observation. We have now included 95%
confidence intervals (CI) for all key comparisons throughout the manuscript. The tables (Tables 2–4)
have been updated to display CI for mean differences, median, correlation coefficients, and regression
coefficients, and the corresponding results have been revised accordingly in the text. These
modifications were implemented in lines 37, 38, 40, 42, 43, 203, 206, 207, 212, 213, 224, 227, 228,
241, 242 and 257.

Comments 4: In the discussion please mention potential confounding by medications and
comorbidities not assessed.
Response 4: We thank the reviewer for this valuable suggestion. We have now expanded the
Discussion section to acknowledge the potential confounding influence of unassessed
medications and comorbidities, including familial hypercholesterolemia, obstructive sleep
apnea, atrial fibrillation, psoriasis, and vitamin D deficiency, as well as the possible effects of
antituberculosis therapy and agents such as doxycycline, azithromycin, and isotretinoin.
This addition appears in the revised manuscript (Discussion section, page 10-11, lines 344-
355).
“A potential limitation of this study is the possibility of residual confounding re-lated to
medications and comorbidities that were not fully assessed. Certain conditions known to
influence hematologic or inflammatory parameters, such as familial hyper-cholesterolemia,
obstructive sleep apnea syndrome, atrial fibrillation, psoriasis, or vitamin D deficiency, may
alter red blood cell morphology and platelet activation independently of coronary
atherosclerosis. In addition, specific medications not evaluated in our cohort, including
antituberculosis therapy and agents such as doxycycline, azithromycin, or isotretinoin, have
been reported to affect erythropoiesis, oxidative balance, or platelet indices. Although
patients with major systemic and hematologic disorders were excluded, these unmeasured
confounders may have contributed to subtle variations in RDW and MPV values and should
be considered in future prospective studies.”

Comments 5: Should apply multiple testing correction (Bonferroni or FDR)
Response 5: We thank the reviewer for this valuable methodological recommendation
regarding the application of multiple testing correction (Bonferroni or FDR). Following the
constructive suggestion of another reviewer, the statistical analyses were refined to ensure
full consistency with the study objectives. Specifically, in the correlation analysis, only the
erythrocyte and platelet distribution indices were retained, while in the univariate and
multivariate regression analyses, only the distribution indices and echocardiographic
measurements directly reflecting cardiac function were included. As a result of this
substantial reduction in the number of analyzed variables, the application of multiple testing
correction is no longer considered necessary in the revised version of the manuscript.

Comments 6: Table 4 need to include confidence intervals for all estimates.
Response 6: In accordance with the comment, Table 4 has been revised to include 95%
confidence intervals for all univariate and multivariate regression coefficients. These
additions provide a clearer representation of the precision and robustness of each estimate.
Corresponding updates were also made in the Results section, where all regression
parameters are now reported together with their 95% CI.

Comments 7: The authors should consider reading “Deciphering Abnormal Platelet
Subpopulations in COVID-19, Sepsis and Systemic Lupus Erythematosus through Machine
Learning and Single-Cell Transcriptomics" to provide insights into platelet heterogeneity
that could inform the interpretation of hematological biomarkers in cardiovascular disease.
Response 7: We thank the reviewer for this valuable recommendation. We have carefully
reviewed the article. A short paragraph has been added to the Discussion to acknowledge
recent evidence regarding platelet heterogeneity and its potential implications for the
interpretation of hematological indices in cardiovascular disease. (Discussion section, page
10, lines 334-343).
“Recent evidence has shown that platelet populations are highly heterogeneous in terms of
size, density, and functional reactivity, which may influence the interpretation of
hematological indices such as MPV and PDW in cardiovascular disease. This heteroge-neity
may partly explain the modest and sometimes inconsistent associations between platelet
indices and coronary atherosclerosis severity. In this regard, a recent study by Qiu et al.
identified distinct platelet subpopulations associated with disease severity and clinical
outcomes, suggesting specific mechanisms of platelet disorders in COVID-19, sepsis, and
systemic lupus erythematosus. These findings highlight the bio-logical complexity of
platelet activation pathways and provide a conceptual framework for understanding the
variability of platelet-derived hematological biomarkers in car-diovascular settings.”

Comments 8: Remove redundant keywords.
Response 8: We thank the reviewer for this observation. The keywords “erythrocyte
indices,” “platelet indices,” and “acute coronary syndrome” have been removed as
requested. The remaining keywords have been revised to better reflect the scope and focus
of the study.

Comments 9: Add R² values to scatter plots.
Response 9: We thank the reviewer for this helpful suggestion. The R² values have been
added to the scatter plots. (page 7-8, Figure 2)

Comments 10: Clarify definition of "typical angina".
Response 10: We thank the reviewer for this comment. The definition of “typical angina”
has been clarified in accordance with the ESC criteria, specifying that it refers to chest
discomfort often described as pressure, heaviness, or tightness, provoked by exertion or
emotional stress and relieved by rest or nitroglycerin (Methods section, page 3, lines 114-
117).
“Typical angina was defined according to the European Society of Cardiology criteria as
chest discomfort, often described as pressure, heaviness, or tightness, that is provoked by
exertion or emotional stress and relieved by rest or nitroglycerin within a few minutes.”

Comments 11: Add normal reference ranges for hematological parameters.
Response 11: We thank the reviewer for this helpful suggestion. The normal reference
ranges for hematological indices have been added in the revised manuscript to improve
methodological clarity. These specifications appear in the Methods section, page 3-4, lines
142–144.
“The normal reference ranges for hematological indices were considered as follows: red cell
distribution width 11.5–14.5%, mean corpuscular volume 80–100 fL, platelet distri-bution
width 9–17%, and mean platelet volume 7.5–11.5 fL.”

Reviewer 2 Report

Comments and Suggestions for Authors
  • Inform consent-
    • What were the subjects informed about when signing the informed consent?
      The fact that the research is retrospective does not mean that the subjects whose data from the medical documentation are presented were informed about it. It looks like the subjects have signed the consent that any of their data from the medical documentation can be used for scientific research purposes, which is not in accordance with the ethical norms of COPE
  • Gensini score- reference must be provided
  • Please clarify the rationale for performing parametric statistics in this study. Were data normally distributed? Did you perform any statistical tests to evaluate distribution normality? Which one?
  • distributed? Did you perform any statistical tests to evaluate distribution normality? Which one?

    Table 1 must be improved and clarified regarding title and subtitles. Tables also must be self explanatory y and authors must provide enough details regarding the composition.

    For example

    General characteristic of the study population

    Female sex, % (N)

    81 (195)

    Age, years –  median (IQR)

    84 (46-94)

    Body mass index, kg/m2 – median (IQR)

    24.5 (23.7–29.3)

    Smoking, % (N)

    5 (0.25)

    Glucose, mmol/L, median (27th; 75th)

    5. (4.24; 6.o)

    AST, U/L, mean±SD

    27.5 ± 12.2

    Results are expressed as percentage for…., mean and SD, median and IQR (25th and 75th percentile) depending on data distribution….

  • SI units must be performed as much as possible, for glucose, and lipids as well not just for electrolytes

  • Please omit declaration about "intensity of significance", because this is useless: * p<0.05 statistically significant, ** p<0.01 very statistically significant, *** p<0.001 extremely 214
    statistically significant
    • The interpretation of the correlation coefficient results is completely wrong because if P<0.05, it means that we can interpret the correlation, and only based on the coefficient value can we conclude whether there is any correlation at all or whether it is weak to moderate, (r=0.25-0.5), moderate to good (r=0.5-0.75), very good to excellent (0.75-1). If the coefficient is below 0.25, there is no correlation. And this is also visible from the images of the correlation lines that the authors presented because the points are quite scattered from the line, except in the case of creatinine. Therefore, the results for creatinine are completely unexpected and it is questionable whether they are accurate. Given the "clogged" nature of the points on the line itself, we would expect a significantly higher correlation coefficient
    • Educational material that can help: Udovičić M, Baždarić K, Bilić-Zulle L, Petrovečki M. What we need to know when calculating the coefficient of correlation?. Biochem Med (Zagreb). 2007;17:10-15

Author Response

Dear Reviewer, (PLEASE SEE THE ATTACHMENT)

We thank you for your insights for this manuscript. We are sure that these suggestions improve our article. 

Point-by-point response to Comments and Suggestions for Authors

Comments 1: Inform consent-What were the subjects informed about when signing the
informed consent? The fact that the research is retrospective does not mean that the subjects
whose data from the medical documentation are presented were informed about it. It looks
like the subjects have signed the consent that any of their data from the medical
documentation can be used for scientific research purposes, which is not in accordance with
the ethical norms of COPE.
Response 1: We sincerely thank the reviewer for this valuable remark regarding the
informed consent procedure and the ethical compliance of our retrospective study. We fully
acknowledge the importance of transparency and adherence to COPE ethical standards.
We would like to clarify that our study was conducted in full accordance with Romanian
and European data protection and research ethics regulations. The research protocol
received ethical approval both from the Hospital Ethics Committee and from the Ethics
Committee of the “Victor Babeș” University of Medicine and Pharmacy, Timișoara.
According to Law No. 190/2018 – on measures for the implementation of the GDPR in
Romania, which complements the EU General Data Protection Regulation (GDPR):
• The processing of sensitive data (such as medical information) for scientific research
purposes is permitted without the explicit consent of the data subject, provided there
is a legal basis (e.g., public interest, academic or medical research activity) and the
data are anonymized or pseudonymized;
• Such research activities must be approved and supervised by the Ethics Committee
of the medical or university institution;
In our study, all data were anonymized prior to analysis, and no identifiable patient
information was accessed or disclosed. Therefore, the research fully complies with national
legislation, GDPR requirements, and COPE ethical principles.

Comments 2: Gensini score- reference must be provided
Response 2: We thank the reviewer for this valuable remark. We have now added an
appropriate and up-to-date reference describing the Gensini score calculation method. The
following citation has been included in the revised manuscript: Rampidis, G.P.; Benetos, G.;
Benz, D.C.; Giannopoulos, A.A.; Buechel, R.R. A Guide for Gensini Score Calculation.
Atherosclerosis 2019, 287, 181–183, doi:10.1016/j.atherosclerosis.2019.05.012.

Comments 3: Please clarify the rationale for performing parametric statistics in this study.
Were data normally distributed? Did you perform any statistical tests to evaluate
distribution normality? Which one?
Response 3: We thank the reviewer for this important observation regarding the statistical
methodology. Following this valuable comment, we carefully rechecked the distribution of
all continuous variables and performed the appropriate statistical tests accordingly.
Normality was assessed using the Shapiro–Wilk test. Variables that met the normality
assumption were expressed as mean ± standard deviation and analyzed using parametric
tests (Student’s or Welch’s t-test). Variables that did not follow a normal distribution were
expressed as median (interquartile range) and analyzed using nonparametric tests, such as
the Mann–Whitney U test for group comparisons and Spearman’s rank correlation
coefficient for associations.
We have revised and reformulated the corresponding paragraph in the Materials and
Methods section to clarify the statistical approach. (line 176-188). In addition, the tables and
corresponding results have been revised accordingly (Table 1, Table 2, page 6-7; line 201-204,
223-229, Table 4, Figure 2)

Comments 4: Table 1 must be improved and clarified regarding title and subtitles. Tables
also must be self explanatory and authors must provide enough details regarding the
composition.
Response 4: We thank the reviewer for this helpful observation. In response, Table 1 and the
subsequent tables have been revised and clarified. The titles and subtitles were adjusted for
better readability, and additional details regarding the composition of each variable were
added. Moreover, the tables were adjusted to clearly distinguish between parametric and
nonparametric variables, with the corresponding statistical tests and data presentation
(mean ± SD or median [IQR, 25th–75th percentile]) specified accordingly.

Comments 5: SI units must be performed as much as possible, for glucose, and lipids as well
not just for electrolytes.
Response 5: We thank the reviewer for this valuable observation regarding the use of SI
units. We fully acknowledge the importance of standardization, however, in our country,
laboratory results for glucose and lipid parameters are routinely reported in mg/dL, in
accordance with national laboratory practice and clinical guidelines.

Comments 6: Please omit declaration about "intensity of significance", because this is
useless: * p<0.05 – statistically significant, ** p<0.01 – very statistically significant, *** p<0.001
– extremely statistically significant
Response 6: We thank the reviewer for this helpful remark. In accordance with the
suggestion, we have removed the declaration regarding the intensity of significance from the
manuscript and tables. Statistical significance is now reported uniformly as p-values,
without additional qualitative labels.

Comments 7: The interpretation of the correlation coefficient results is completely wrong
because if P<0.05, it means that we can interpret the correlation, and only based on the
coefficient value can we conclude whether there is any correlation at all or whether it is
weak to moderate, (r=0.25-0.5), moderate to good (r=0.5-0.75), very good to excellent (0.75-1).
If the coefficient is below 0.25, there is no correlation. And this is also visible from the images
of the correlation lines that the authors presented because the points are quite scattered from
the line, except in the case of creatinine. Therefore, the results for creatinine are completely
unexpected and it is questionable whether they are accurate. Given the "clogged" nature of
the points on the line itself, we would expect a significantly higher correlation coefficient
Educational material that can help: Udovičić M, Baždarić K, Bilić-Zulle L, Petrovečki M.
What we need to know when calculating the coefficient of correlation?. Biochem Med
(Zagreb). 2007;17:10-15
Response 7: We sincerely thank the reviewer for this detailed and very helpful comment
regarding the interpretation of the correlation coefficients. We have carefully reviewed and
consulted the material suggested by the reviewer to ensure the correct interpretation of
correlation strength. Following this recommendation, we have revised the correlation
analysis by excluding all parameters that were not part of the primary objective, specifically
the red cell and platelet distribution indices. The Results and Discussion sections have been
updated accordingly.
“A statistically significant but weak positive correlation (Table 3) was observed between the
Gensini score and RDW (rho=0.28, 95%CI: 0.16-0.39, p==0.001) (Figure 2.a) , indicating that
increased coronary lesion complexity is associated with greater anisocytosis. Although some
comparisons reached statistical significance (p < 0.05), no meaningful correlation was
observed between RDW–SD (rho=0.14, 95%CI: 0.01-0.26, p=0.02), MPV (rho=0.15, 95%CI:
0.02-0.27, p=0.01), and the Gensini score, as the corre-sponding Spearman’s rho values were
below 0.25, indicating only very weak monotonic associations.” -RESULTS- 3.2 Correlation
analyses (page 7 line 223-229)

4. Response to Comments on the Quality of English Language
Point 1: The English could be improved to more clearly express the research.
Response 1: We thank the reviewer for this valuable observation. The entire manuscript has
been carefully revised for English language clarity and fluency. Grammar, terminology, and
sentence structure were refined to improve readability and to ensure that the research is
expressed more clearly and accurately

Reviewer 3 Report

Comments and Suggestions for Authors

Please see the attached file. Thank you.

Author Response

Dear Reviewer, (PLEASE SEE THE ATTACHMENT)

We thank you for your insights for this manuscript. We are sure that these suggestions improve our article. 

 Point-by-point response to Comments and Suggestions for Authors

Comments 1: Suggest changing L-53
「Mean Platelet Volume, Platelet Distribution Width, and Red Cell Distribution Width are…」
to 「Mean Platelet Volume (MPV) ,
Platelet Distribution Width (PDW), and Red Cell Distribution Width (RDW) are…]
Response 1: We thank the reviewer for this constructive suggestion. The requested
modifications have been implemented accordingly and can be found in lines 52–53 of the
revised manuscript.

Comments 2: Suggest adding a reference after the description of L-102 「(L-95〜102)
Although the SYNTAX score…. research on prognostic biomarkers.
Response 2: A relevant reference has been added after the description of the SYNTAX score
to support the statement and provide additional context regarding prognostic biomarkers.
The bibliographic references 16–19 have been added, and the paragraph between lines 91
and 104 has been adjusted to better align with the study methodology.

Comments 3: What was the age of the patients included in the study?
Response 3: We thank the reviewer for the question. Descriptive statistical data related to
age have been included in the revised manuscript.
“The median age of the patients included in the study was 64 years (IQR 56–70 years), with a
minimum age of 33 years and a maximum of 87 years.” Page 5, line 193-194 and Table 1.

Comments 4: What were the age differences between the low score and high score groups?
Response 4: We thank the reviewer for the question. Descriptive statistical data related to
age have been included in the revised manuscript.
“The median age was significantly higher in the High Score group compared to the Low
Score group (65 [56–71] vs. 62 [54–69] years, p = 0.04; Mann–Whitney U test). The Hodges–
Lehmann estimate of the median difference was 3.0 years (95% CI: –6.0 to -0.2), indicating
that older age was associated with more severe coronary atherosclerosis.” This information
has been added to the Results section (page 5, line 201-204 and Table 1) of the revised
manuscript.

Comments 5: Suggest changing L-154〜155「such as the proximal left anterior descending
and left circumflex arteries…」 to 「such as the proximal left anterior
descending (LAD) and left circumflex arteries (LCX)…].
Response 5: We thank the reviewer for this helpful suggestion. The text has been revised
accordingly to specify the abbreviations for the coronary arteries. The phrase now reads:
“such as the proximal left anterior descending (LAD) and left circumflex arteries (LCX)”, as
indicated in lines 166–167 of the revised manuscript.

Comments 6: Suggest changing L-168「To assess … RDW-SD, MCV, PDW…」 to 「To
assess … RDW-SD, Mean corpuscular volume (MCV), PDW…」
Response 6: We thank the reviewer for this precise observation. The sentence has been
modified to include the full term before the abbreviation, now reading: “To assess … RDW-
SD, mean corpuscular volume (MCV), PDW…”, as shown in line 186 of the revised
manuscript.

Comments 7: Suggest changing L-179「with a mean ejection fraction of 48.96 (± 7.12). The
baseline …」 to 「with a mean left ventricular ejection fraction of 48.96 (±
7.12)%. The baseline …」
Response 7: We thank the reviewer for this valuable suggestion. The phrase has been
revised to specify the parameter more precisely and to include the measurement unit, now
reading: “with a mean left ventricular ejection fraction of 48.96% (± 7.12).” The modification
appears in line 197 of the revised manuscript.

Comments 8: Suggest changing L-186「total cholesterol, LDL cholesterol,…」 to 「 total
cholesterol, Low-density lipoprotein cholesterol,…」
Response 8: We thank the reviewer for this helpful suggestion. The text has been revised to
include the full term before the abbreviation, now reading: “total cholesterol, Low-density
lipoprotein cholesterol, …”, as indicated in line 209 of the revised manuscript.

Comments 9: Suggest changing Table 1 and Table 2「Ejection Fraction」 to 「Left
Ventricular Ejection Fraction」
Response 9: We thank the reviewer for this useful observation. The variable name has been
revised in both Table 1 and Table 2 to read “Left Ventricular Ejection Fraction” instead of
“Ejection Fraction,” ensuring terminological accuracy and consistency throughout the
manuscript.

Comments 10: It is recommended that the full term be written out be the first time it appears
in the article. Suggest changing L-210「between the Gensini score …, TAPSE…」
to 「between the Gensini score …tricuspid annular plane systolic excursion
(TAPSE) …」
Response 10: We thank the reviewer for this helpful recommendation. In line with the
reviewer’s feedback (comments 13–14), TAPSE has been excluded from the manuscript’s text
and remains only in Tables 1 and 2.

Comments 11: Suggest ensuring consistency between the text (such as L-210) and Table 1〜3
regarding blood glucose values ( fasting vs. non-fasting).
Response 11: We thank the reviewer for this important observation. All reported blood
glucose values now refer explicitly to fasting glucose measurements, ensuring uniform
terminology throughout the manuscript. In line with the reviewer’s feedback (comments 13–
14) blood glucose has been excluded from the manuscript’s text and remains only in Tables 1
and 2.

Comments 12: L-92〜L-95:「the primary objective… investigate the relationship between
the severity of coronary lesions…and elevated RDW and PDW values. 」 As the
main purpose of the article is to investigate the relationship between the severity
of coronary lesions and RDW / PDW values, it may be helpful if figure 2 also
presents this relationship.
Response 12: We sincerely thank the reviewer for this valuable suggestion. To better
illustrate the main objective of the study, a new Figure 2 has been added to the revised
manuscript. This figure presents the relationship between the Gensini score and PDW
values, using scatter plots.

Comments 13: There not very consistent between the variables analyzed statistically (L-171
〜L-172:「Multivariate linear regression analysis was employed to identify 2
independent predictors of the Gensini Score 」) and the variables intended to be
examined in the study (L-92〜L-95:「the primary objective… investigate the
relationship between the severity of coronary lesions…and elevated RDW and
PDW values. 」). For instance, the multivariate linear regression analysis
includes variables, such as gender(male), smoking, hypertension, diabetes,
creatinine, blood glucose, GTO, GTP, ejection fraction, diastolic dysfunction,
wall motion abnormality and TAPSE, that were not part of the original study
objectives.
Response 13: We sincerely thank the reviewer for this valuable observation and for
highlighting the inconsistency between the initial study objectives and the variables
included in the multivariate regression analysis. Following this recommendation, the
statistical model has been revised and adjusted. Irrelevant parameters such as gender,
smoking, hypertension, diabetes, serum creatinine, AST, ALT, blood glucose, and TAPSE
have been excluded. The revised multivariate model now focuses on the primary variables
of interest, accompanied by parameters that directly assess cardiac function, ensuring a
more coherent alignment with the study’s main objectives. Corresponding modifications
have been made in Table 4 and lines 185-188, 249-258 of the revised manuscript.
Comments 14: It may be appropriate to consider removing the left ventricular ejection
fraction, TAPSE, fasting blood glucose, AST, ALT, and creatinine variables in table 3, as
it does not appear directly relevant to the study objectives.

Response 14: We thank the reviewer for this valuable suggestion. Following the
recommendation, Left Ventricular Ejection Fraction, TAPSE, Fasting Blood Glucose, AST,
ALT, and Creatinine were removed from the correlation analysis to maintain only the
parameters relevant to the main objectives of the study. This adjustment ensured a clearer
focus on the hematological variables directly related to coronary lesion complexity. Table 3
has been revised, and the paragraph corresponding to lines 220–226 has been removed to
improve the clarity and consistency of the results section.
“Furthermore, the Gensini score demonstrated a significant positive correlation with serum
creatinine levels (r = 0.196, 95%CI: 0.07-0.31,p = 0.002) (Figure 2.d), highlighting an
association between more extensive coronary artery disease and impaired renal func-tion. In
contrast, a significant negative correlation was identified between the Gensini score and left
ventricular ejection fraction (r = –0.222, 95%CI: -0.33 to -0.09 p < 0.001), indicating that
increased coronary complexity is linked to reduced systolic function”-removed

Comments 15: For the same reason as stated in comment 16, it is recommended to remove
figure 2 d.
Response 15: We thank the reviewer for this observation. Figure 2d has been removed, as
recommended, and replaced with the new figure suggested in comment 12, which more
clearly illustrates the relationship between PDW and the Gensini score.

Comments 16: Suggest changing table 4「with a mean ejection fraction of 48.96 (± 7.12). The
baseline …」 to 「with a mean left ventricular ejection fraction of 48.96 (±
7.12)%. The baseline ……」
Response 16: We thank the reviewer for this valuable suggestion and acknowledge the
consistency with comment 7. The term has been revised to “left ventricular ejection fraction”
as recommended. This modification has been applied not only to Table 4 but also
consistently across all tables in the manuscript to ensure uniform terminology and clarity.

Comments 17: May I ask if GTO refers to glutamyl oxaloacetic transaminase? If so it is
recommended to use「AST」.
Response 17: We thank the reviewer for this clarification. Yes, GTO refers to glutamyl
oxaloacetic transaminase. Following the recommendation, the term has been replaced with
“AST” throughout the manuscript to ensure standardized terminology and consistency with
current biochemical nomenclature.

Comments 18: May I ask if GTP refers to alanine aminotransferase? If so it is recommended
to use「ALT」.
Response 18: We thank the reviewer for this clarification. Yes, GTP refers to alanine
aminotransferase. Following the reviewer’s recommendation, the term has been replaced
with “ALT” throughout the manuscript to ensure accuracy and consistency with standard
biochemical terminology.

Comments 19: It may be helpful to include information on research ethics, for example, by
providing the IRB approval number.
Response 19: We thank the reviewer for this suggestion. When the abstract and manuscript
were submitted, the ethical approval obtained from both the hospital and the university was
already included as a separate document.

Comments 20: In this study, the degree of stenosis in each coronary artery was assessed via
coronary angiography, and the overall severity of coronary artery stenosis was
quantified using the Gensini score. The degree of coronary artery stenosis was
independently evaluated by two cardiologists. It is recommended to report the
inter-rater agreement and clarify how the final assessment of coronary artery
stenosis was determined.
Comments 21: In general, the angiography system itself is equipped with built-in software
that can estimate the degree of coronary artery stenosis, therefore, providing details on the
angiographic view(s) used for the estimation, or clarifying whether the
estimation was based on a combination of multiple views, would help enhance
the credibility of the data.
Comments 22: It is recommended to provide information on the precision and/or accuracy
of the instruments used to obtain the data in this study.
Response 20&21&22: We thank the reviewer for this insightful comment. The sentence “Two
interventional cardiologists independently visualized and interpreted the results” has been
revised to provide a clearer and more accurate description of the evaluation process. It now
reads: “The coronary angiography procedures were jointly performed by two experienced
interventional cardiologists. Each atherosclerotic plaque was carefully analyzed in multiple
angiographic projections and, when appropriate, quantitatively assessed using the
angiograph’s integrated measurement software. This collaborative approach ensured
accurate evaluation of lesion severity and minimized potential errors associated with purely
visual assessment.” The revised version appears in lines 151-157 of the manuscript.

Comments 23: Suggest changing table 4「RCDW」 to 「RDW」.
Response 23: We thank the reviewer for this helpful observation. The typographical error
has been corrected, and “RCDW” has been replaced with “RDW” in Table 4 to ensure
consistency and accuracy across the manuscript.

Comments 24: L-262〜L-264:「Tzur et al. [21] demonstrated… PDW levels …a higher risk of
90-day mortality. 」 The results of this study differ from Tzur et al, and this study found
no correlation between Gensini score and PDW (p=0.133). It may be worthwhile to address
this point in the discussion.
Response 24: We thank the reviewer for this helpful observation.
“However, in our study, which focused specifically on patients with chronic coronary artery
disease, PDW did not show a predictive value for disease severity. It should be noted that
short-term outcomes, such as 90-day mortality, were not evaluated in our cohort, which may
partly explain the lack of prognostic significance. Further prospective studies are warranted
to explore the potential predictive role of PDW in different clinical settings and over longer
follow-up periods” was added page 10, line 328-333.

Comments 25: L-276〜L-277:「Our findings demonstrated a positive correlation between
the severity of coronary artery lesions and ….RDW and MPV… 」 Although
the results of this study are consistent with previous literature or the initial
hypothesis (showing that the Gensini score/ disease severity score is associated
with RDW and MPV, respectively), it is recommended to clearly state that the
values of RDW and MPV values in this study remained within the normal range,
and to address this point in the discussion.
Response 25: We thank the reviewer for this valuable comment. We have clarified in the
Discussion section that, although RDW showed a positive correlation with the Gensini score
and both RDW and MPV values were higher in the high-score group, these parameters
remained within their respective normal reference ranges in our study population. This
point has now been explicitly addressed and discussed in the revised manuscript. Page 9-10,
line 289-298

4. Response to Comments on the Quality of English Language
Point 1: The English could be improved to more clearly express the research.
Response 1: We thank the reviewer for this valuable observation. The entire manuscript has
been carefully revised for English language clarity and fluency. Grammar, terminology, and
sentence structure were refined to improve readability and to ensure that the research is
expressed more clearly and accurately

Round 2

Reviewer 1 Report

Comments and Suggestions for Authors

The authors revised the manuscript accordingly. 

Author Response

We thank the reviewer for their prior feedback, it brought significant improvements to the manuscript.

Reviewer 2 Report

Comments and Suggestions for Authors

Si-units must be provided. This is an international open access journal and it will be visible worldwide not just in your country.

Author Response

We thank the reviewer for their second round of feedback. All these suggestions improved the quality of the article.

Comment 1: Si-units must be provided. This is an international open access journal and it will be visible worldwide not just in your country.

Response 1: We fully agree that all laboratory parameters should be expressed in SI units, as this ensures international comparability and clarity for a global readership. Accordingly, all biochemical values (including fasting blood glucose, total cholesterol, LDL, triglycerides, and creatinine) have been converted and are now reported in SI units (mmol/L or µmol/L) throughout the manuscript, including in Tables 1 and 2.

Reviewer 3 Report

Comments and Suggestions for Authors

Author Response

We thank the reviewer for their second round of feedback. All these suggestions improved the quality of the article.

Comment 1: Please check if there are any typos Line 186: Mean corpuscular volume (MPV), and PDW with the...

Response 1: We thank the reviewer for this observation. The sentence has been revised accordingly for clarity and consistency.

“To assess the independent association of RDW, RDW-SD, Mean corpuscular volume (MCV), MPV and PDW with the Gensini score, potential confounding variables…..” -line 186-187

Comment 2:

The confidence interval (CI) values in Table 2 are suggested to be verified by a statistical expert, as categorical variable (nominal scale) is not appropriate for calculating CI. For example, in the case of gender, where male is coded as 1 and female as 0, these codes are merely categorical lables, and the resulting CI values are meaningless. If the intention is to present the percentage distribution of gender, CI can be used for that purpose. The CI values for gender appears to have been calculated based on a nominal scale.

Response 2:

We thank the reviewer for this valuable remark. We have rechecked the statistical analysis and recalculated the appropriate tests using the Chi-square test for categorical variables.

The 95% confidence intervals now refer to the difference between proportions between the two study groups. Corresponding modifications have been made in the manuscript (line 207) and in Table 2 for the following variables: Male, Smoker, Diabetes mellitus, Hypertension, Diastolic dysfunction, and Wall motion abnormality.

“Differences between proportions were assessed using the Chi-square test (or Fisher’s exact test when appropriate).”- was added line 183-184 in Materials and Methods-Statistical analyses.